# Hybrid-Aligned Fibers of Electrospun Gelatin with Antibiotic and Polycaprolactone Composite Membranes as an In Vitro Drug Delivery System to Assess the Potential Repair Capacity of Damaged Cornea

**DOI:** 10.3390/polym16040448

**Published:** 2024-02-06

**Authors:** Yi-Hsin Shao, Ssu-Meng Huang, Shih-Ming Liu, Jian-Chih Chen, Wen-Cheng Chen

**Affiliations:** 1Advanced Medical Devices and Composites Laboratory, Department of Fiber and Composite Materials, Feng Chia University, Taichung 407, Taiwan; angelshao2001@gmail.com (Y.-H.S.); dream161619192020@gmail.com (S.-M.H.); 0203home@gmail.com (S.-M.L.); 2Department of Orthopedics, Faculty of Medical School, College of Medicine, Kaohsiung Medical University, Kaohsiung 807, Taiwan; 3Department of Orthopedics, Kaohsiung Medical University Hospital, Kaohsiung 807, Taiwan; 4Department of Fragrance and Cosmetic Science, College of Pharmacy, Kaohsiung Medical University, Kaohsiung 807, Taiwan; 5Dental Medical Devices and Materials Research Center, College of Dental Medicine, Kaohsiung Medical University, Kaohsiung 807, Taiwan

**Keywords:** electrospinning, gelatin, polycaprolactone, membrane, antibiotic, corneal damage, biocompatibility

## Abstract

The cornea lacks the ability to repair itself and must rely on transplantation to repair damaged tissue. Therefore, creating alternative therapies using dressing membranes based on tissue engineering concepts to repair corneal damage before failure has become a major research goal. Themost outstanding features that are important in reconstructing a damaged cornea are the mechanical strength and transparency of the membrane, which are the most important standard considerations. In addition, preventing infection is an important issue, especially in corneal endothelial healing processes. The purpose of this study was to produce aligned fibers via electrospinning technology using gelatin (Gel) composite polycaprolactone (PCL) as an optimal transport and antibiotic release membrane. The aim of the composite membrane is to achieve good tenacity, transparency, antibacterial properties, and in vitro biocompatibility. Results showed that the Gel and PCL composite membranes with the same electrospinning flow rate had the best transparency. The Gel impregnated with gentamicin antibiotic in composite membranes subsequently exhibited transparency and enhanced mechanical properties provided by PCL and could sustainably release the antibiotic for 48 h, achieving good antibacterial effects without causing cytotoxicity. This newly developed membrane has the advantage of preventing epidermal tissue infection during clinical operations and is expected to be used in the reconstruction of damaged cornea in the future.

## 1. Introduction

In corneal transplant surgery, donated corneas or amniotic membranes are often relied upon as graft material. Long-distance transportation, vibration, and temperature changes can reduce the corneal endothelial cells of the donated cornea, resulting in unstable corneal quality. Amniotic membrane has multiple functions such as antibacterial, anti-protease, and anti-inflammation mechanisms; the inhibition of new blood vessels; and scar removal. However, it needs to be obtained from amniote embryos, the thickness of the material is not fixed, and the amniotic membrane degrades faster than wounds can heal [1,2,3]. Therefore, artificial cornea preparation technology was developed in recent years to improve the quality of the cornea and solve problems such as insufficient cornea quantity.

In recent years, many artificial corneas have been developed through tissue engineering technology. Tissue engineering consists of three main elements: cells, scaffolds, and signals such as growth factors. Among them, scaffolds are the most important part. They are not only required to provide cell attachment and proliferation but must also be able to withstand the tension caused by high intraocular pressure and eye movements [4,5]. Many methods can be used to prepare hydrogel bio-scaffolds with a large number of interconnected pores, such as solvent casting and salt leashing, thermal phase separation, freeze-drying, and electrospinning. Hydrogel membrane produced through electrospinning technology as a nanofiber scaffold naturally has the advantages of high specific surface area, good tension, flexible and thin film, and high porosity [6,7,8]. In addition to simulating the structure of the extracellular matrix [9], the aligned fibers of electrospun hydrogel membranes can serve as biological scaffolds to guide cell attachment and proliferation, thus performing better than fibrous membranes with randomly arranged fibers [10,11,12,13].

During the selection of scaffold materials based on artificial corneal membranes, emphasis should be placed on improving their mechanical strength, transparency, biocompatibility, and biodegradability. According to the literature [14], natural polymers, such as collagen, silk fibroin, gelatin (Gel), and chitosan, can change cell behavior and have good transparency, but they still have problems such as fast degradation and poor mechanical properties. Synthetic polymers can enhance their mechanical properties but are less transparent. Some studies pointed out that blending natural and synthetic polymers can simultaneously achieve sufficient mechanical strength, excellent transparency, and biodegradability [15,16,17]. Cross-linked Gel membranes hinder transparency. However, while Gel membranes without cross-linking have good transparency, they remain unusable for clinical applications due to their lack of mechanical strength, leading to rapid dissolution. The tensile strength of the membrane can be enhanced by adding synthetic polymers such as polylactic acid, polyvinyl alcohol, and polycaprolactone (PCL) [18,19]. PCL is a hydrophobic semi-crystalline linear polymer with appropriate mechanical properties and slow biodegradability [20]. However, its hydrophobicity limits the effects on cell attachment, proliferation, and migration [21]. In addition, PCL has poor transparency. When used to repair corneal damage, patients cannot wait for the membrane to biodegrade to restore vision [22]. Therefore, this study hypothesizes that using electrospinning technology to hybridize PCL and Gel into fibrous membranes can improve the low mechanical strength, transparency, hydrophilicity, and cell attachment capabilities of the membrane.

Infection often tends to occur after artificial corneal membrane surgery [23]. Although infection can be prevented through eye drops, the patient’s compliance is uncontrollable, resulting in unsatisfactory treatment results. Therefore, the antibiotic–impregnated membrane can not only improve the above shortcomings and increase the contact time between the drug and the wound but also enable the sustained and slow release of antibiotics, thereby improving the therapeutic effect [24,25,26]. Electrospun antibiotic–impregnated Gel and PCL composite fiber membranes were not studied in corneal repair. Therefore, an experiment was designed in the present study to hybridize electrospun antibiotic–impregnated Gel and PCL antibacterial composite fiber membranes. This experiment will maximize the advantages of aligned/oriented fibers; explore their physical and chemical properties, antibacterial properties, and biocompatibility; and find the best conditions for antibiotic–impregnation Gel composite PCL, aiming to obtain antibacterial composite membranes with good transparency and mechanical strength.

## 2. Materials and Methods

### 2.1. Materials

The raw materials used in this study included polycaprolactone (PCL, M_n_: 80,000 g/mole), gelatin (type B, average molar mass of 40,000–50,000 g/mole), and N,N-dimethylformamide (DMF, purity > 99.0%) purchased from Sigma-Aldrich^®^ (St. Louis, MO, USA). Other chemicals included dichloromethane (DCM, CH_2_Cl_2_, purity ≥ 99.5%, Mw: 84.93 g/mole, Avantor^®^, Radnor, PA, USA), citric acid 1-hydrate (PanReac AppliChem, Barcelona, Spain), and gentamicin (Genta, Siu Guan Chemical Industrial Co., Chiayi, Taiwan).

### 2.2. Preparation of Functional PCL/Gel Composite Membrane

#### 2.2.1. Experimental Solution for Electrospun Processes

The PCL patches were added to DCM/DMF at a 1:3 volume ratio, and the sample was stirred for 4 h at room temperature by controlling the speed at 400 rpm to prepare a 10 wt.% PCL solution. After the Gel was added to 40 wt.% acetic acid solution, 1 mL of gentamicin (Genta) was dissolved to prepare concentrations of 0, 20, and 40 mg/mL and then stirred for 3 h at a controlled temperature of 40 °C. The 30 wt.% Gel solutions that formed contained different concentrations of the antibiotic Genta.

#### 2.2.2. Electrospun Composite Membranes

The 10 wt.% PCL solution (Figure 1a) and the 30 wt.% Gel solution (Figure 1b) were placed in separate syringes, which were then set on the micro-pusher. Both syringes were connected to a high-voltage power supply. The output voltage was adjusted to 16 kV, and then different flow rates were set to control the co-electrospinning ratios of PCL and Gel (Figure 1c) to 10:0, 7:3, 5:5, and 0:10 (abbreviated as 10P, 7P3G, 5P5G, and 10G groups, respectively). The collection distance of the electrospinning was adjusted to 15 cm, and the linear fiber acquisition speed of the drum collector with a diameter of 10 cm was 10.7 m/s. After 6–10 h of collection, a composite membrane with aligned-fiber membrane was obtained.

### 2.3. Preparation of Functional PCL/Gel Composite Membrane

#### 2.3.1. Observation by Optical Microscope

Optical microscopy (OM) analysis (CK, Olympus, Tokyo, Japan) can initially determine whether the parameter settings during sample preparation were under optimal conditions. The electrospun fibers were collected on a glass slide for 10 to 20 s and displayed on OM to observe whether the image exhibited an optimal condition, i.e., whether the fiber morphology was smooth and uniform or in a droplet-like state. Qualitative characterization of cytotoxicity was also performed through OM.

#### 2.3.2. Observation through Scanning Electron Microscope

In this study, fiber membrane samples were cut into dimensions of 0.5 × 0.5 cm and platinum-coated for 30 s. Then, the fiber distribution and the surface morphology were observed through a scanning electron microscope (SEM) (S-3400N, Hitachi, Tokyo, Japan).

#### 2.3.3. Transparency Test

The electrospun membranes collected under different electrospinning conditions were cut into 2.0 × 1.0 cm size and soaked in phosphate-buffered saline (PBS) solution with pH ~7.4 and commercial physiological normal saline (NS) at room temperature for 15 min and then placed on cardboard with printed fonts. Then, the sharpness of the fonts in each set of membranes was compared as a qualitative result of the transparency tests.

The quantitative transparency test uses UV-Vis (UV1800, Shimadzu, Kyoto, Japan) to measure the absorbance (A), and then it uses the Beer–Lambert law to convert the absorbance (A) into the transmittance (T%) formula to express the quantitative results of transparency [14].
$$\mathrm{Beer}–\mathrm{Lambert}\;\mathrm{law}:A=\log\;\frac{1}{T}$$
$$T\%=10^{-A}\times 100$$

#### 2.3.4. Water Wettability

A fixed amount of 50 μL deionized water was dropped on the surface of the 2 × 2 cm composite membranes, and the shape of the water droplets was observed using a contact drop angle measuring instrument (CAM121, Creating Nano Technologies Inc., Tainan, Taiwan) equipped with a charge-coupled device (CCD). Accordingly, the hydrophilic and hydrophobic properties of the membrane surface were obtained.

#### 2.3.5. Tensile Strength Measurement

Dumbbell-shaped samples were prepared according to ASTM D882-02 [27] specifications, which is the Standard Test Method for Tensile Properties of Thin Plastic Sheeting. After the thickness of the sample was recorded, the sample was tested with a universal material testing machine (HT-2402, Hongda, Taichung City, Taiwan) at a tensile speed of 2 mm/min. Tensile strength was recorded until the specimen broke. Tensile strength studies were performed on dry and wet composite samples to investigate changes in mechanical properties upon wetting. The wetted samples were soaked in double-distilled water for 10 min and then subjected to tensile strength testing.

#### 2.3.6. Fourier-Transform Infrared Spectroscopy

Fourier-transform infrared spectroscopy (FTIR) (Nicolet iS5, Thermo Fisher Scientific, Waltham, MA, USA)-measured spectra can confirm whether new functional groups are generated, which is an important basis for judging whether additives are chemically bonded with polymers. Nanofibrous membranes were partially opaque and, therefore, tested using an FTIR-attenuated total reflectance technique.

#### 2.3.7. Degradation Rate of Weight Loss In Vitro

A 2 × 2 cm cut square sample was placed in a 15 mL centrifuge tube, 5 mL PBS was added, and the solution was stored in a 37 °C constant-temperature water tank. The sample weight at a fixed time point was measured, and the changes in sample weight were measured through the following formula [28]:$$Weight\;loss\left(\%\right)=\frac{W_{0}-W_{x}}{W_{0}}\times 100\%$$

### 2.4. Antimicrobial Tests

A semi-qualitative antibacterial test was performed using the agar diffusion test, using tryptic soy broth as the bacterial nutritious medium. The antibacterial activity of membranes against two pathogenic bacteria (Gram-positive and -negative) was studied using the American Institute of Clinical and Laboratory Standards disc diffusion method. *Staphylococcus aureus* (*S. aureus*; ATCC number: 25923) and *Escherichia coli* (*E. coli*; ATCC number: 10798) were cultured in commercial Luria-Bertani broth. Samples with a diameter of 6 mm were first prepared and then subjected to UV sterilization for 24 h, placed on bacteria-coated plates, and incubated at 37 °C for 24 h. Then, the inhibition zone was observed and recorded to determine the antimicrobial efficiency.

The antibacterial quantitative part was tested using the broth dilution method. A sample weighing 0.01 g was soaked in 1 mL of bacterial suspension with an optical density (OD_595_) of 0.2 using an ELISA reader (SPECTROstar Nano, BMG Labtech, Offenburg, Germany) and cultured with bacteria at 37 °C. The control group, the DMSO (Sigma-Aldrich^®^, St. Louis, MO, USA) positive group, and the sterilization experimental group were cultured for 1, 4, 8, 24, and 48 h. At the set time, 100 µL of bacterial liquid was taken to measure its OD_595_ value, and the bacterial load was calculated.

### 2.5. Cytotoxicity In Vitro

#### Cytotoxicity

The cell line selected for the cytotoxicity test was mouse fibroblasts (L929, National Institutes of Health, Miaoli, Taiwan), and the test was conducted in accordance with the specifications of ISO 10993-5 [29]. The medium used was the widely deployed Minimum Essential Medium α, which contains 10% horse serum, penicillin and streptomycin antibiotics, and sodium bicarbonate. The samples were sterilized by UV-LED irradiation for 8 h, and the results showed that the samples were sterilized. Sample extracts need to be prepared first to quantify cytotoxicity. The sample in this experiment is a porous membrane with a thickness of less than 0.5 mm; thus, the extraction ratio of the sample in the culture medium is 6 cm^2^/1 mL. Four groups were processed. The extracts of PCL/Gel composite membranes were the experimental groups. The others included the control group, which was a cell cultured normally in a culture medium. The positive group revealed the form of cell death by culturing with the medium in a concentration of 15 vol.% dimethyl sulfoxide (DMSO, Sigma-Aldrich, St. Louis, MO, USA). The negative group used high-density polyethylene to confirm whether sterilization was effective.

A total of 100 µL of the culture medium and L929 cells with a cell concentration of 1 × 10^4^ cells/well were placed in a 96-well micro-plate and cultured in a 37 °C 5% CO_2_ incubator for 24 h. Then, the culture medium was removed, and the extract was added for culture. After 24 h, the extraction solution was removed, 100 μL/well of new cell culture medium with 50 μL/well of XTT cell proliferation assay kit (Biological Industries Ltd., Beit Haemek, Israel) was added, mixed evenly, and cultured for 4 another h. Afterward, an ELISA reader was used to measure the cell viability of OD_492_, based on which OD_492_ absorbance was directly proportional to cell viability.

For the qualitative test of cytotoxicity, the sample extract group is the same as the quantitative cytotoxicity. The extracts with a culture medium of 1000 µL and L929 cells with a cell concentration of 1 × 10^5^ cells/well were placed in a 48-well micro-plate and cultured in a 37 °C 5% CO_2_ incubator for 24 h. The sample was placed under an inverted microscope (IVM-3AFL, SAGE VISION Co., Ltd., New Taipei City, Taiwan) to observe cell morphology.

### 2.6. Statistical Analysis

The statistical software IBM SPSS Statistics 20 (IBM Co., Armonk, NY, USA) was used for statistical analysis. For data of tensile strength, analysis of variation (ANOVA) was used to explore whether statistically significant differences existed in the stress and strain results of PCL/Gel composite membranes, and Tukey’s test was performed as pairwise post hoc testing.

## 3. Results

The OM image morphology of various fiber types for each group of controlled electrospinning is shown in Figure 2a. The upper row shows images of electrospun PCL fibers with different volume ratios of PCL and Gel, and the lower row shows electrospun Gel fibers. The PCL and Gel fibers maintained a good shape, and no obvious beads and droplets can be observed. Figure 2b shows the fiber morphology and structure of co-electrospun PCL/Gel composites on the membrane surface with different PCL/Gel ratios. Each group of fibers is aligned along the same direction, i.e., well-oriented fibers. In addition, PCL fibers (10P, average diameter 0.44 ± 0.11 μm; *n* = 50) are thicker than Gel fibers (10G, average diameter 0.25 ± 0.05 μm; *n* = 50). Therefore, a reasonable deduction is that the thicker fibers are PCL and the thinner fibers are gelatin in 7P3G and 5P5G. In the electrospinning thickness measurement of each group (*n* = 50), the film thicknesses of 10P, 7P3G, 5P5G, and 10G were 8.94 ± 1.91, 16.98 ± 3.82, 10.71 ± 2.43, and 2.88 ± 0.67 mm, respectively. The results show that the higher the PCL content, the thicker the fiber diameter and the thicker the electrospun membrane.

The qualitative test of transparency of each group of electrospun membranes after soaking in PBS and NS medium at room temperature for 15 min is shown in Figure 3a. The pure PCL of 10P had the lowest transparency, whether it was soaked in PBS or NS. However, the transparency improved as the volume of Gel in PCL/Gel composites increased. The Gel-only group (10G) was not cross-linked to maintain transparency and facile drug release; thus, the membrane would be dissolved immediately upon contact with the solution, resulting in a maximum Gel contained in the composite membrane of 50% (5P5G). Therefore, the qualitative transparency analysis results showed that 5P5G had the best transparency. The same results were also obtained in quantitative transparency tests (Figure 3b,c). The 5P5G group had the best transparency after soaking in the medium. In addition, the transparency of the membranes soaked in the NS was higher than that of the PBS.

Figure 4 shows the water contact angles of each group of electrospun membranes at different time points using CCD-captured images at 0, 5, and 10 s. In group 10P, the membrane surface remained hydrophobic, with contact angles of 115.9° and 111.8° at 0 and 10 s, respectively. Although the contact angles of the 7P3G and 5P5G groups were hydrophobic, having contact angles of 111° and 116° at 0 s, the contact angles decreased significantly after 5 s of contact. After being in contact with water for 10 s, 7P3G became hydrophilic and dropped to 81.60°, and the water contact angle of 5P5G even dropped to an extremely hydrophilic 25.37°. The results show that the hydrophilicity of the membrane can be improved by forming an electrospun composite membrane through hybrid Gel and PCL [30]. The hydrophilic/hydrophobic properties of artificial corneas play a key role in corneal endothelial tissue engineering scaffolds. In addition to being highly water-absorbent and promoting cell adhesion, the hydrophilic membrane can also help the penetration of nutrients such as glucose and the adsorption of proteins [31]. As for the 10G group, because it had not been cross-linked, the Gel dissolved when it came into contact with water, so its water contact angle could not be measured. In this study, 5P5G had the smallest contact angle, showing that it has optimal hydrophilicity to enhance the metabolic behavior of cells.

The typical stress–strain curves of dry and wet samples of each group of electrospun membranes after tensile testing are shown in Figure 5. As expected, the pure Gel fiber of 10G had the worst mechanical properties, while the pure PCL fiber of 10P had the best mechanical strength of the dry sample test (Figure 5a). Therefore, the composite properties, i.e., tensile strength and strain, of the 7P3G and 5P5G groups of hydride Gel and PCL were better than those of the pure Gel group.

For use in corneal surgery, artificial corneal devices should provide sufficient strength to withstand stresses associated with intraocular pressures of 10–21 mmHg, i.e., 1.3–2.8 kPa [32]. Table 1 shows the mechanical properties of each group of electrospun membranes. According to statistical analysis, the mechanical properties of 10P, 7P3G, and 5P5G are significantly different from those of the 10G pure gelatin group (*p* < 0.05), and no difference was found between the two composite groups of 7P3G and 5P5G (*p* > 0.05). The above results show that composite membranes incorporating PCL with Gel can indeed largely improve the mechanical strength compared with pure Gel as a single component. Since pure gelatin (10G) dissolved immediately after soaking, while pure PCL (10P) had no transparency after immersion, the wetted 10G and 10P samples were not tested for tensile strength. When wetted samples of PCL/Gel were subjected to tensile testing, the stresses of 7P3G and 5P5G were significantly reduced by 98% compared to dry testing (Figure 5b). This significant decrease in strength was caused by gelatin dissolution, leading to the destabilization of the entangled structure within the membrane. However, its stress value can still withstand the pressure required for an intraocular pressure of 1.3–2.8 kPa (Table 2).

Although Gel has good transparency, it lacks mechanical strength, while PCL has excellent mechanical properties but poor transparency. The membrane of the 5P5G composite had the best transparency, improved mechanical properties, and the best hydrophilicity among PCL/Gel composites. Therefore, the subsequent treatments were performed by adding different concentrations of antibiotic Genta (5P5G_Genta20 and 5P5G_Genta40) to the 5P5G composition in electrospun Gel fibers.

The detailed construction images of composite membranes are shown in Figure 6, presenting the fiber morphology and aligned arrangement structure after Genta was added to Gel for electrospun membranes. The addition of Genta to the Gel did not affect the observed Gel morphology and fiber arrangement.

Figure 7 shows the qualitative transparency test of composite membranes soaked in PBS and NS for 15 min. Optical imaging showed that the transparency of the composites with Genta added to 5P5G_Genta20 and 5P5G_Genta40 was indistinguishable from that of 5P5G without Genta, meaning that the addition of Genta did not affect the transparency.

Figure 8 shows the infrared spectrum of each electrospun composite membrane. Pure material spectra for PCL, Gel, and Genta are also disclosed. The absorption of PCL at 2936 and 2861 cm^−1^ is the characteristic stretching band of H–C–H, 1721 cm^−1^ is the stretching band of C=O, and 1292 cm^−1^ is C–O or C–H. The bands at 1239 and 1168 cm^−1^ are (C–O–C) [33]. The bands of amide I (C=O; C–N) at 1630 cm^−1^, amide II (N–H) at 1524 cm^−1^, and amide III (C–N) at 1238 cm^−1^ can be observed in the Gel curve [34,35]. In the Genta spectrum, individual bands of stretched O–H at 3615 cm^−1^, stretched N–H at 1679 cm^−1^, and bent O–H at 838 cm^−1^ can be observed [36].

The spectra of 5P5G, 5P5G_Genta20, and 5P5G_Genta40, referring to the H–C–H, C=O, C–O, C–H, and C–O–C absorption bands of PCL, and amide I (C=O; C–N) and amide II (N–H) absorption bands of Gel show that Genta’s bands overlapped with those of PCL and Gel. Thus, Genta could not be verified. An antibacterial test was subsequently performed to confirm the presence of gentamicin on the membrane.

After the composite membrane was soaked in PBS for 5 min, the greatest weight was achieved, indicating that the membrane was saturated with water. Therefore, the degradation test was started after each composite was soaked in the PBS for 5 min. As can be seen from Figure 9, as the soaking time increases, the uncross-linked Gel in the membrane begins to degrade, causing the weight of each group to slowly decrease over time. In addition, the antibiotic Genta in the Gel fiber would not affect the original degradation trend. Thus, the electrospun hybrid fiber formed by the combination of Gel and PCL has the ability to slow down the degradation time of Gel fiber alone. With the release of Genta, the slow degradation time of Gel in PCL/Gel composites can be extended to at least 28 days, making this type of composite membrane a candidate material for corneal cell repair scaffolds.

Figure 10 shows the qualitative antibacterial activity of each group of membranes against *S. aureus* and *E. coli* after 24 h of culture. The 5P5G membrane without Genta showed no inhibition zone, indicating that the 5P5G group did not have antibacterial ability. The 5P5G_Genta20 and 5P5G_Genta40 groups had excellent antibacterial effects against *S. aureus* and *E. coli*. In addition, as the concentration of Genta in the Gel increased, the inhibition zone tended to expand, especially for *E. coli.* This result demonstrated that the electrospun membrane successfully contained different concentrations of Genta in the Gel.

Figure 11a,b show the quantitative antibacterial test against *S. aureus* and *E. coli* after each group of membranes was cultured for 1, 4, 8, 24, and 48 h. The control group is pure bacterial liquid, and DMSO is the positive control group that causes bacterial death. The OD values of 5P5G_Genta20 and 5P5G_Genta40 after 4 h of culture were lower than those of the control group. Even at 24 and 48 h, their OD values were significantly lower than those of the control group, thus proving the consistent and sustainable antibacterial ability of 5P5G_Genta20 and 5P5G_Genta40. In addition, the OD value of 5P5G_Genta40 was lower than that of 5P5G_Genta20, confirming that 5P5G_Genta40 has better antibacterial efficiency. At the same time, Genta’s antibacterial effect against *E. coli* is worse than that against *S. aureus*, and the membrane was proven to continuously release drugs for at least 48 h.

Figure 12a shows the quantitative testing of cytotoxicity. The cell survival rate of each electrospun membrane was higher than 70%, indicating that the extracts of each group were non-cytotoxic. Figure 12b shows the qualitative testing of the cytotoxicity of each electrospun membrane. The results showed that the cells in both the control group and the negative control group were spindle-shaped, indicating that the cells were healthy and adhered well. The results verified that the sterilization status of the experimental group was good. In the positive control group, cells shrank and took on a spherical shape, indicating cell death. A comparison of the cell membranes of each group showed that the cells all grew well and did not induce any cytotoxicity.

## 4. Conclusions

In summary, the fiber morphology of each electrospun membrane was continuous, uniform, and well-aligned, and no electrospun beads appeared. The transparency and water contact angle test results showed that the PCL fibrous membrane had poor transparency and exhibited hydrophobicity. However, as the volume ratio of added Gel increased, its transparency and hydrophilicity also improved. The 5P5G hybrid fiber membrane had the best transparency and hydrophilicity. In terms of tensile strength, after Gel fiber and PCL fiber were hybridized, the mechanical strength of the composite membrane could be greatly improved compared with that of pure Gel fibrous membrane. After different antibiotic concentrations of Genta to the Gel fibers were added, the transparency did not change, and the fibers achieved antimicrobial properties against *S. aureus* and *E. coli.* The 5P5G_Genta40 experimental group had the best antibacterial effect and remained effective after 48 h of testing. Degradation rate analysis showed that hybridizing Gel fiber and PCL fiber into a composite membrane through electrospinning can greatly delay the degradation rate of Gel-only fiber. In this experiment, the antibiotic gentamicin-impregnated gelatin fiber hybrid polycaprolactone fiber composite membrane was successfully prepared using co-electrospinning technology. The 5P5G_Genta40 group can achieve transparency, mechanical strength, hydrophilicity, bacteriostasis, and delayed degradation without inducing cytotoxicity. This newly developed hybrid PCL/Gel membrane is expected to be used in corneal clinical trials in the future.

## Figures and Tables

**Figure 1 polymers-16-00448-f001:**
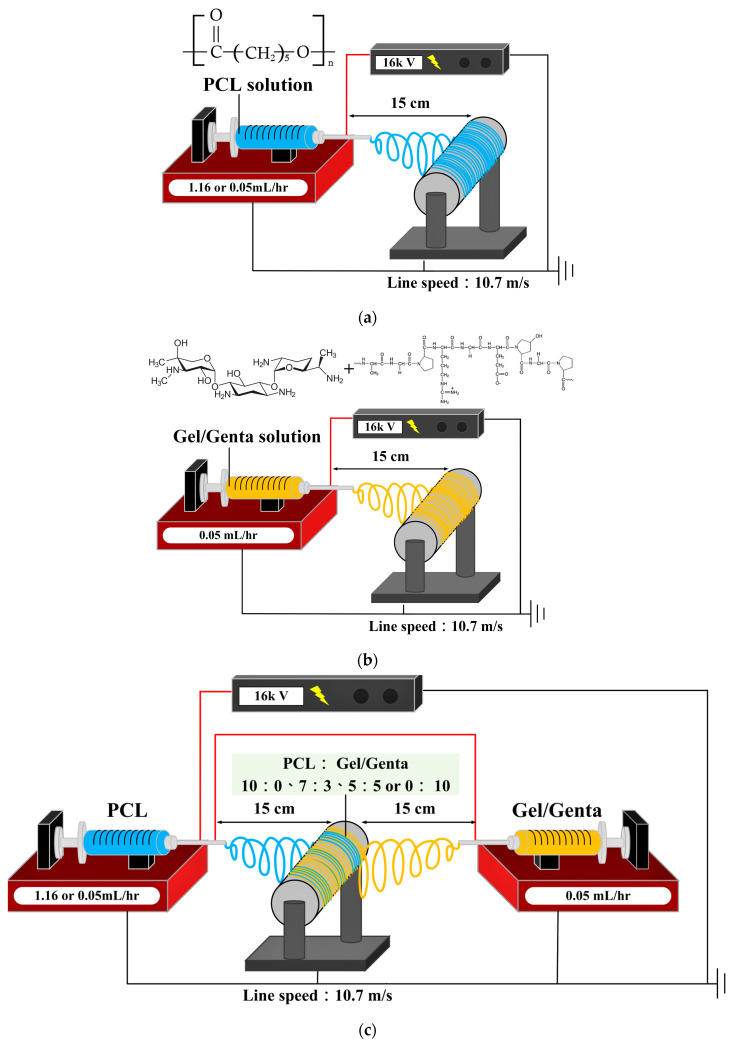
Simulation diagrams and parameters of different composite membranes used in this experiment: (**a**) electrospinning of aligned fibers with 10 wt.% PCL solution; (**b**) electrospinning of aligned fibers with Genta-impregnated 30 wt.% Gel solution; (**c**) co-electrospinning to prepare PCL composite Genta-impregnated Gel hybrid fiber membranes.

**Figure 2 polymers-16-00448-f002:**
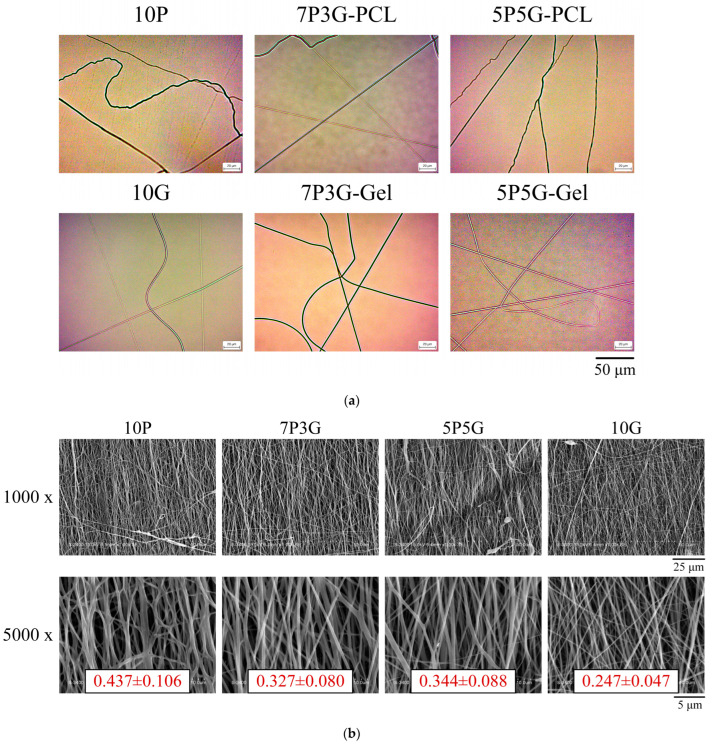
Electrospun fiber morphologies of PCL/Gel composite membranes prepared with different volume ratios: (**a**) the OM images (the upper row shows the PCL fibers, and the lower row reveals the Gel fibers) and (**b**) the SEM images with different magnification.

**Figure 3 polymers-16-00448-f003:**
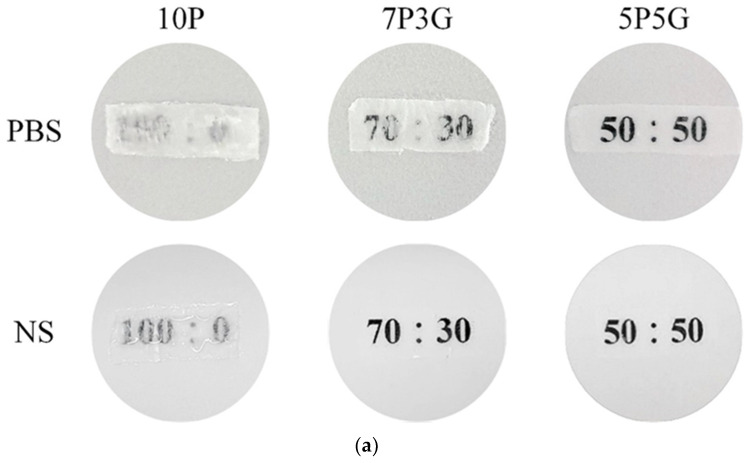
Qualitative transparency of different PCL/Gel ratios in composite membranes soaked in PBS and NS for 15 min at room temperature (**a**); quantitative transparency of the immersed PCL/Gel samples in PBS (**b**) and NS for 15 min (**c**) (*n* = 6).

**Figure 4 polymers-16-00448-f004:**
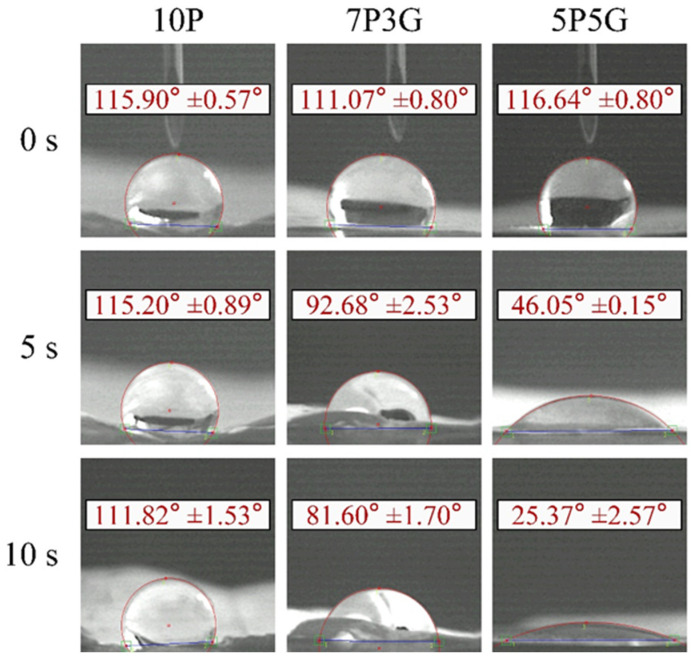
Images of water contact angle changes of PCL/Gel composite membranes with different electrospun volume ratios at different time points (*n* = 3).

**Figure 5 polymers-16-00448-f005:**
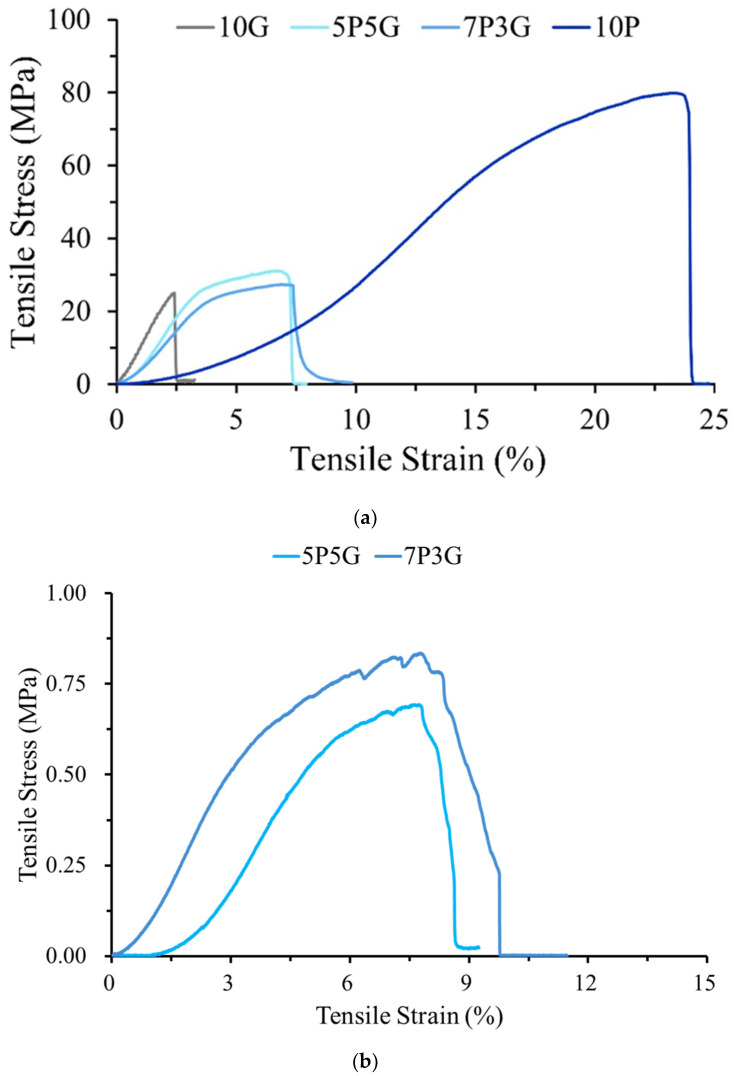
Typical stress–strain curves of (**a**) dry sample and (**b**) wetted sample of PCL/Gel composite membranes with different PCL composite Gel proportions stretched along the fiber alignment direction, and the wetted sample was soaked in double-distilled water for 10 min.

**Figure 6 polymers-16-00448-f006:**
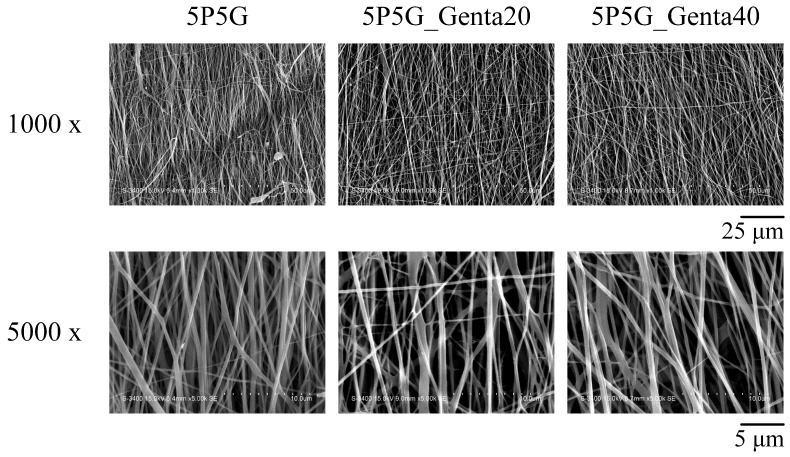
SEM images of PCL/Gel composite membranes (5P5G without incorporated Genta) containing different antibiotic concentrations.

**Figure 7 polymers-16-00448-f007:**
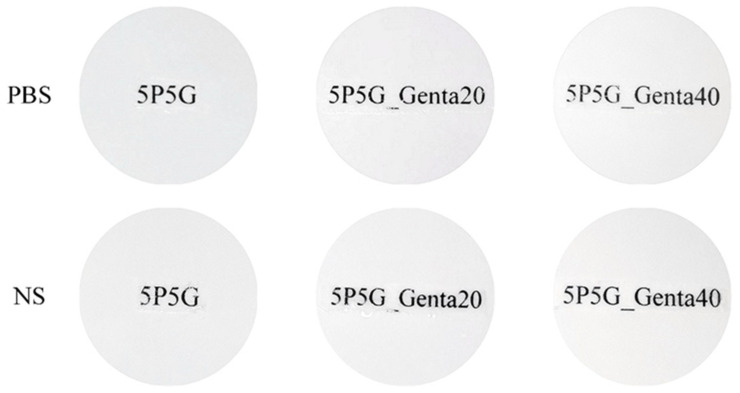
Qualitative transparency of PCL/Gel composite membranes containing different Genta concentrations soaked in PBS and NS for 15 min at room temperature.

**Figure 8 polymers-16-00448-f008:**
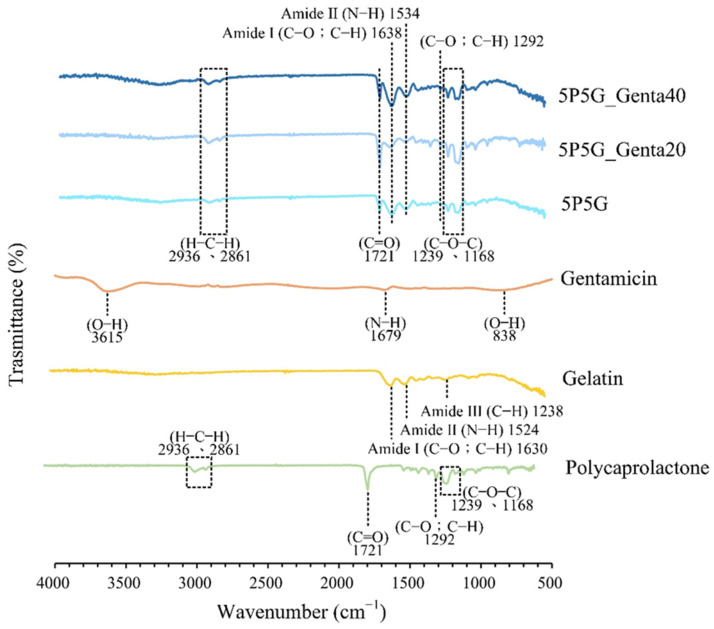
ATR-FTIR spectra of PCL, Gel, Genta raw materials, and PCL/Gel composite membranes with different Genta concentrations.

**Figure 9 polymers-16-00448-f009:**
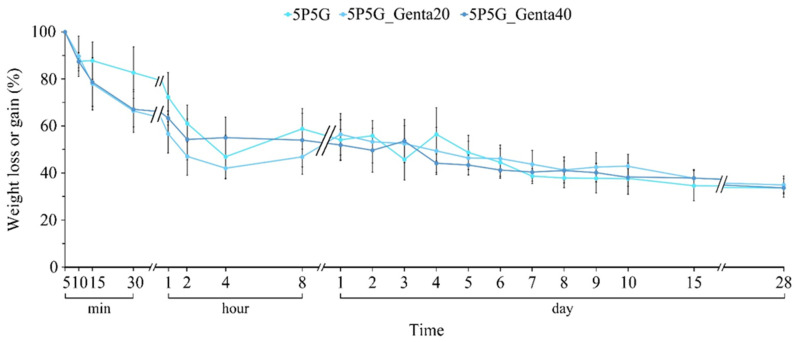
In vitro degradation analysis of 5P5G without antibiotic compared with PCL/Gel composite membranes containing different gentamicin concentrations (*n* = 6).

**Figure 10 polymers-16-00448-f010:**
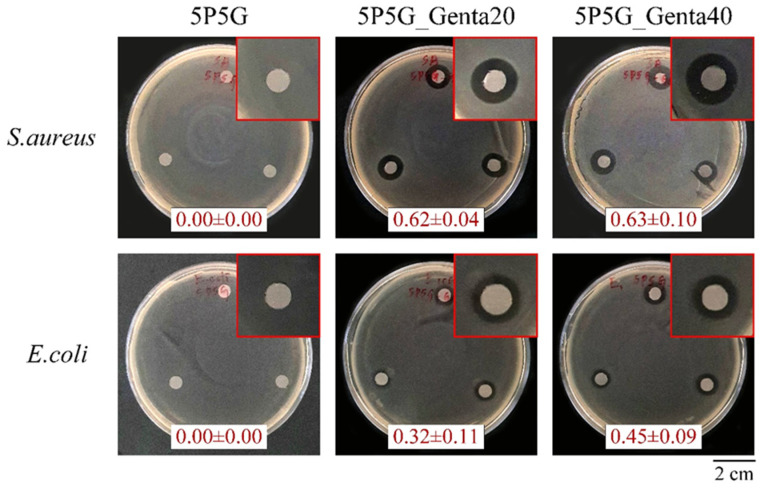
Qualitative test of the antibacterial activity of PCL/Gel composite membranes with different Genta concentrations against *S. aureus* and *E. coli* after incubation for 24 h (*n* = 3).

**Figure 11 polymers-16-00448-f011:**
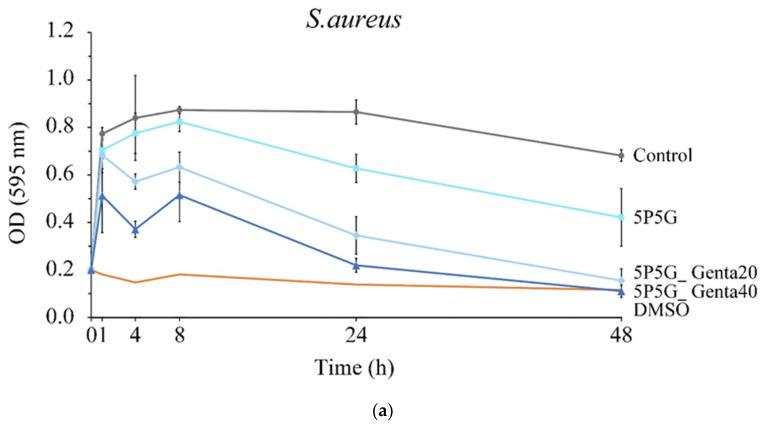
Quantitative antibacterial activity of PCL/Gel composite membranes containing different Genta concentrations against (**a**) *S. aureus* and (**b**) *E. coli* after culture for 1, 4, 8, 24, and 48 h (*n* = 3).

**Figure 12 polymers-16-00448-f012:**
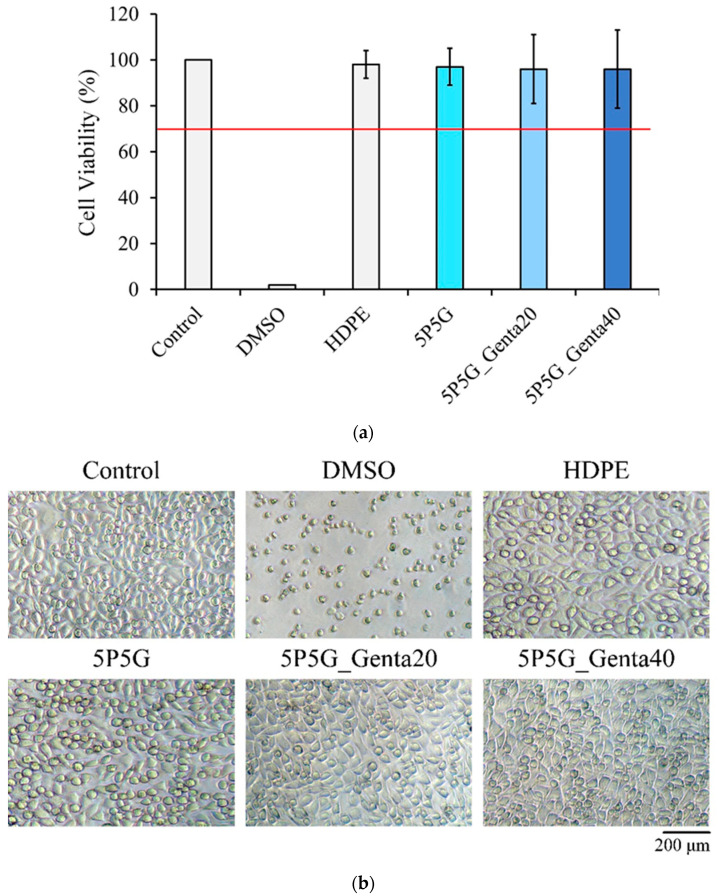
(**a**) Cytotoxicity quantification (*n* = 6) and (**b**) cell morphology observation after culturing PCL/Gel composite membrane containing different Genta concentrations with L929 cells for 1 day. The red line represents 70% viability, above which material extracts are considered non-cytotoxicity.

**Table 1 polymers-16-00448-t001:** Tensile stress and strain values of dry PCL/Gel composite membranes with different PCL composite Gel proportions stretched along the fiber alignment direction (*n* = 10).

Designated Groups	Tensile Stress (MPa) (Dry Samples)	Strain (%) (Dry Samples)
10P	83.06 ± 13.40	24.15 ± 2.74
7P3G	28.07 ± 1.31 *	7.08 ± 0.75 *
5P5G	32.12 ± 3.74 *	6.85 ± 0.77 *
10G	24.94 ± 6.11 *	2.63 ± 0.38 *

* shows significant differences between ANOVA and multiple pairwise comparisons, *p* < 0.05.

**Table 2 polymers-16-00448-t002:** Tensile stress and strain values of wetted samples of 7P3G and 5P5G stretched along the fiber alignment direction after soaking in double-distilled water for 10 min (*n* = 10).

Groups	Tensile Stress (MPa) (Wetted Samples)	Strain (%) (Wetted Samples)
7P3G	0.83 ± 0.12	7.21 ± 1.08
5P5G	0.75 ± 0.11	7.86 ± 0.24

## Data Availability

Data are contained within the article.

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
