# Peer review of "Hybrid-Aligned Fibers of Electrospun Gelatin with Antibiotic and Polycaprolactone Composite Membranes as an In Vitro Drug Delivery System to Assess the Potential Repair Capacity of Damaged Cornea"

_polymers, 2024, doi:10.3390/polym16040448_

Round 1

Reviewer 1 Report

Comments and Suggestions for Authors

The manuscript contains the results of a scientific study on the development of a new biocompatible material that can be used for corneal prosthetics. The introduction can be shortened by removing general phrases and formulating the problems to be solved more specifically.

The methods and results with discussion are excellent in describing the particular properties of the resulting materials.

But the main question is what happens to the gelatin when it comes into contact with a liquid, and how can this affect the mechanical properties? If the gelatin dissolves, this would explain the excellent transparency in the 5P5G group. However, the strength of the material should change due to the loss of some of the gelatin fibers. The study of mechanical strength is performed on dry samples. And it is not clear how the mechanical properties of the material will change after wetting.

Second, the main method of corneal fixation during surgery today is suturing with a 10/0 diameter suture. Therefore, investigating the strength of material retention and cutting with sutures of different diameters could also be included in the study design. The use of fiber alignment during electrospinning suggests in advance the different force required to cut through the surgical suture in the longitudinal and transverse directions.

The manuscript should include the exact thickness of the material being tested for each test.

The size of the fragments examined should be reported correctly: either the area in square millimeters or the linear dimensions in millimeters (lines 138, 154, 171).

The presence of polysaccharides in the PBS solution is puzzling because they should not be there by default (line 259).

All of the above questions require a reasoned response.

Author Response

We appreciate the efforts of the Reviewers and Editor, which have helped us to greatly improve this manuscript. I have considered “all of the Reviewers’ comments of Polymers-2848697”, which was re-submitted in Polymers. All reviewer comments have been addressed, red marked, or marked with the “track changes” in the revised manuscript, and all remarked sentences have been also rephrased on the highlights of Editor checked. We hope that the revised modifications are acceptable.

Detailed responses to reviewers’ comments

-----------------------------------------------------------------------------------------------------------------------------

Reviewer 1

The manuscript contains the results of a scientific study on the development of a new biocompatible material that can be used for corneal prosthetics. The introduction can be shortened by removing general phrases and formulating the problems to be solved more specifically.

The methods and results with discussion are excellent in describing the particular properties of the resulting materials.

Q1: But the main question is what happens to the gelatin when it comes into contact with a liquid, and how can this affect the mechanical properties? If the gelatin dissolves, this would explain the excellent transparency in the 5P5G group. However, the strength of the material should change due to the loss of some of the gelatin fibers. The study of mechanical strength is performed on dry samples. And it is not clear how the mechanical properties of the material will change after wetting.

Response: I want to thank the Reviewers for their time and suggestions. We have heeled the suggestion and made the proper changes in Results (red marked sections on pages 10–12, revision). The results for the wetted specimens after tensile testing are included in the newly added Figure 5b and Table 2 (revised version). The explanation “Tensile strength studies were performed on dry and wet composite samples to investigate changes in mechanical properties upon wetting. The wetted samples were soaked in double-distilled water for 10 min and then subjected to tensile strength testing.” added in section 2.3.5. (Page 5, revision).

Q2: Second, the main method of corneal fixation during surgery today is suturing with a 10/0 diameter suture. Therefore, investigating the strength of material retention and cutting with sutures of different diameters could also be included in the study design. The use of fiber alignment during electrospinning suggests in advance the different force required to cut through the surgical suture in the longitudinal and transverse directions.

Response: Thank you for your important feedback, which we will include in future research.

Q3: The manuscript should include the exact thickness of the material being tested for each test.

Response: Thanks for your comment. We have adopted the recommendations and provided the exact thickness of the membranes being tested for each group (track changes on page 7, revision).

Q4: The size of the fragments examined should be reported correctly: either the area in square millimeters or the linear dimensions in millimeters (lines 138, 154, 171).

Response: Thank you for the remarks. We have heeded the reviewer’s comments and corrected the unit.

Q5: The presence of polysaccharides in the PBS solution is puzzling because they should not be there by default (line 259). All of the above questions require a reasoned response.

Response: Thanks for your comment and appreciate the detailed inspection. We heeded the reviewer’s comments and removed inappropriate sentences (track changes on page 8, revision).

Reviewer 2 Report

Comments and Suggestions for Authors

Hybrid-aligned fibers of electrospun gelatin with antibiotic and polycaprolactone composite membranes as an in vitro drug delivery system to assess the potential repair capacity of damaged cornea 

The study presents a novel and effective method for producing membranes for alternative therapy of replacing damaged cornea. The innovative approach to producing membranes made from collagen and gelatin, which are transparent and also have mechanical strength. In addition, preventing infection using antibiotics present in its own material represents a meaningful advancement in the field.

The paper’s strengths lie in its detailed experimental methods, a number of measurements of material properties, antimicrobial tests, and also testing of cytotoxicity. These aspects will definitely interest other researchers in this area.

However the way the authors showed some of methods is a bit confusing in parts. It is not clear from the methods how the samples were sterilized for the cytotoxicity assays and whether this sterilization method could have affected the properties of the membranes.

Moreover, in Figure 2a, how do you explain that cytotoxicity reached a variance greater than 100%?

How did you ensure that only living cells are in Figure 2b? Do you assess cell viability from the images, e.g. using the live/dead assay?

However, this does not detract from the quality and significance of the research as a whole. The findings are solid and the study advances our knowledge of how to make corneal replacements, which could be of great importance for regenerative medicine in the future.

Overall, the paper is a valuable contribution due to its contribution, scientific thoroughness and clarity of presentation, although some elements of the results could be improved. Given its potential impact and value to other researchers, I recommend its acceptance.

Author Response

We appreciate the efforts of the Reviewers and Editor, which have helped us to greatly improve this manuscript. I have considered “all of the Reviewers’ comments of Polymers-2848697”, which was re-submitted in Polymers. All reviewer comments have been addressed, red marked, or marked with the “track changes” in the revised manuscript, and all remarked sentences have been also rephrased on the highlights of Editor checked. We hope that the revised modifications are acceptable.

Detailed responses to reviewers’ comments

------------------------------------------------------------------------------------------

Reviewer 2

The study presents a novel and effective method for producing membranes for alternative therapy of replacing damaged cornea. The innovative approach to producing membranes made from collagen and gelatin, which are transparent and also have mechanical strength. In addition, preventing infection using antibiotics present in its own material represents a meaningful advancement in the field.

The paper’s strengths lie in its detailed experimental methods, a number of measurements of material properties, antimicrobial tests, and also testing of cytotoxicity. These aspects will definitely interest other researchers in this area.

However the way the authors showed some of methods is a bit confusing in parts. It is not clear from the methods how the samples were sterilized for the cytotoxicity assays and whether this sterilization method could have affected the properties of the membranes.

Response: I want to thank the Reviewers for their time and suggestions that made this study more complete. Samples were UV-LED sterilized for 8 h. The relevant description has been added to the revisoin (section 2.5.1, page 6, revised edition). In addition, According to our previous research results, ultraviolet sterilization will not change fiber morphology.

Chen, W.-C.; Ko, C.-Y.; Chang, K.-C.; Chen, C.-H. Influences of processing and sterilizing strategies on reduced silver nanoparticles in poly(vinyl alcohol) electrospun membranes: Optimization and preservation of antibacterial activity. Mater. Chem. Phys. 2020, 254, 123300, doi:https://doi.org/10.1016/j.matchemphys.2020.123300.

Moreover, in Figure 2a, how do you explain that cytotoxicity reached a variance greater than 100%?

Response: Thanks for your comment. The determination of the XTT kit is based on the extracellular reduction of XTT by NADH produced in mitochondria through electron transport across the plasma membrane and electron mediators. Use the ELISA reader to measure the OD value and calculate it relative to the average value of the control group. If the number of viable cells is greater than that of the control group, the cell activity will be greater than 100%. I hope this explanation can be accepted by you.

How did you ensure that only living cells are in Figure 2b? Do you assess cell viability from the images, e.g. using the live/dead assay?

Response: By observing the morphology of L929 cells, spindle cells showed good cell growth and good viability, while round atrophy indicated cell apoptosis. During the experiment, only the cells in the DMSO-positive control group showed round shrinkage, while the cells in other groups grew well. Thanks for your suggestion, the Live/Dead assay can be used to assess cell viability via fluorescence microscopy. In the future, we will include reference materials related to in–situ equipment construction analysis.

However, this does not detract from the quality and significance of the research as a whole. The findings are solid and the study advances our knowledge of how to make corneal replacements, which could be of great importance for regenerative medicine in the future.

Overall, the paper is a valuable contribution due to its contribution, scientific thoroughness and clarity of presentation, although some elements of the results could be improved. Given its potential impact and value to other researchers, I recommend its acceptance.